# The Association between Infectious Mononucleosis and Cancer: A Cohort Study of 24,190 Outpatients in Germany

**DOI:** 10.3390/cancers14235837

**Published:** 2022-11-26

**Authors:** Christoph Roderburg, Sarah Krieg, Andreas Krieg, Tom Luedde, Karel Kostev, Sven H. Loosen

**Affiliations:** 1Department for Gastroenterology, Hepatology and Infectious Diseases, University Hospital Duesseldorf, Medical Faculty of Heinrich Heine University Duesseldorf, 40225 Duesseldorf, Germany; 2Department of Surgery (A), University Hospital Duesseldorf, Medical Faculty of Heinrich Heine University Duesseldorf, 40225 Duesseldorf, Germany; 3Epidemiology, IQVIA, 60549 Frankfurt, Germany

**Keywords:** EBV, Epstein–Barr-Virus, infectious mononucleosis, lymphoma, cancer, hematopoietic tissue

## Abstract

**Simple Summary:**

Cancer is one of the leading causes of death worldwide. Besides genetic risk factors and non-communicable diseases, chronic infections, including Epstein–Barr virus (EBV) infection, have been identified as promoters of cancer. The aim of our study was to investigate the association between infectious mononucleosis, the clinical manifestation of EBV infection, and cancer development in a real cohort of outpatients in Germany. Patients diagnosed with infectious mononucleosis were found to have an increased incidence of tumors of the hematopoietic and lymphoid tissues and a strong tendency to have a positive association with prostate cancer. Together, our results support the evidence for a role of EBV in the development of various malignancies and may stimulate further research efforts to elucidate the precise involvement of EBV in the carcinogenic process.

**Abstract:**

Background: Cancer represents one of the leading causes of death worldwide. Besides genetic risk factors and non-communicable diseases, chronic infections including Epstein–Barr virus (EBV) infection have been identified as promotors of cancer. In the present manuscript, we evaluated the association between infectious mononucleosis, the clinical manifestation of EBV infection, and cancer development in a real-word cohort of outpatients in Germany. Methods: We used the Disease Analyzer database (IQVIA) and matched a total of 12,095 patients with infectious mononucleosis to a cohort of individuals without infectious mononucleosis based on age, sex, index year, and annual patient consultation frequency between 2000 and 2018. Results: Patients diagnosed with infectious mononucleosis had a cancer incidence of 5.3 cases per 1000 person years versus 4.4 cases per 1000 person years for patients without infectious mononucleosis. In multivariable regression models, infectious mononucleosis showed a trend towards a higher incidence of cancer in general in the age group > 50 years (incidence rate ratio (IRR): 1.32; 95% CI: 1.04–1.67) and among men (IRR: 1.36; 95% CI: 1.07–1.72). Infectious mononucleosis was significantly associated with an increased incidence of tumors of the hematopoietic and lymphoid tissues (IRR: 1.75; 95% CI: 1.22–2.50) and showed a strong trend towards an association with prostate cancer (IRR: 3.09; 95% CI: 1.23–7.76). Conclusion: Infectious mononucleosis is associated with an increased incidence of certain cancer types. The present data from a large real-world cohort support the evidence on a role of EBV in the development of different malignancies and could trigger research efforts to further elucidate its precise involvement in the carcinogenic process.

## 1. Introduction

Cancer is a leading cause of death worldwide, accounting for nearly 10 million deaths in 2020 [1]. The most common types of cancer in 2020 in terms of new cases originate from the breast (2.26 million cases), lung (2.21 million cases), colorectum (1.93 million cases), prostate (1.41 million cases), skin (non-melanoma) (1.20 million cases), and stomach (1.09 million cases) [1]. Tobacco use, alcohol consumption, as well as an unhealthy diet have been identified as major are risk factors for the development of cancer [2]. Besides these non-communicable diseases, recently chronic infections were identified as promotors of cancer in human. Approximately 13% of cancers diagnosed in 2018 globally were attributed to carcinogenic infections, including Helicobacter pylori, human papillomavirus (HPV), hepatitis B virus, hepatitis C virus, and Epstein–Barr virus [3]. 

Epstein–Barr virus (EBV), which affects over 90% of the population in Western countries, belongs to the family of human herpesvirus and is the causative agent of infectious mononucleosis [4]. While the acute infection generally shows a benign course, it is well established that EBV is associated with a number of malignancies in humans [3,5]. Of note, in 1997, EBV was classified as a group 1 carcinogen by the International Agency for Research on Cancer [1,6]. While several studies have shown this association and even deciphered the global epidemiology and geographic distribution of EBV-related cancers [7], limited data from real-world Western cohorts exist. In light of this, we conducted a retrospective cohort study consisting of a large sample of outpatients in Germany with a diagnosis of infectious mononucleosis. The aim of our study was to further elucidate the involvement of EBV in carcinogenesis and to promote potential prevention strategies for cancer development and progression.

## 2. Materials and Methods

### 2.1. Database

The present analysis is based on data from the Disease Analyzer database (IQVIA), which contains drug prescriptions, diagnoses, and basic demographic data obtained directly and in anonymous format from computer systems of general practitioners and specialists in Germany [7]. The database covers approximately 3% of all outpatient practices. It has previously been shown that the panel of included practices is representative for general and specialized practices in Germany [7]. It is not publicly available and access is only granted to employees of IQVIA (Prof. Karel Kostev). The Disease Analyzer database has already been used in previous studies focusing on infectious mononucleosis [8,9]. 

### 2.2. Study Spopulation

This retrospective cohort study included patients (≥14 years) with an initial diagnosis of infectious mononucleosis (ICD-10: B27) in 1274 general practices in Germany between January 2000 and December 2018 (index date; Figure 1). Further inclusion criterium was an observation time of at least 12 months prior to the index date and a follow-up time of at least 12 months after the index date. Patients with cancer (ICD-10: C00-C99), in situ neoplasms (ICD-10: D00-D09), and neoplasms of uncertain or unknown behavior (ICD-10: D37-D48) diagnoses prior to or on index date were excluded. We matched patients with infectious mononucleosis to patients without infectious mononucleosis using propensity scores on the basis of sex, age (±1 year), index year, and annual consultation frequency. Because consultation with general practitioners was much more common among patients with infectious mononucleosis, and higher consultation frequency may increase the likelihood of documentation of other diagnoses, we included consultation frequency per year in the matching. Among individuals without infectious mononucleosis, the index date was the date of a randomly selected visit between January 2000 and December 2018 (Figure 1).

### 2.3. Study Outcomes and Statistical Analyses

The study’s primary outcome was defined as the incidence of cancer (ICD 10: C00-C97) associated with infectious mononucleosis. Differences in sample characteristics between individuals with and without infectious mononucleosis were assessed by the Wilcoxon signed-rank test for continuous age, the Stuart-Maxwell test for categorical age, and the McNemar test for sex and comorbidities including diabetes mellitus (ICD-10: E10-E14), thyroid gland disorders (ICD-10: E00-E07), obesity (ICD-10: E66), diseases of esophagus, stomach and duodenum (ICD-10: K20-K31), diseases of liver (ICD-10: K70-K77), chronic bronchitis and COPD (ICD-10: J42-J44). Multivariable Poisson regression models were used to study the association between infectious mononucleosis and cancer adjusted for these comorbidities. The models evaluating the association between infectious mononucleosis and cancer in general (Table 1), were performed separately for women and men as well as five age groups. The multivariable Poisson regression models regarding the different cancer sites (including lip, oral cavity and pharynx (ICD 10: C00-C14), digestive organs (ICD 10: C15-C26), respiratory organs (ICD 10: C30-C39), skin (ICD 10: C43, C44), breast (ICD 10: C50), female genital organs (ICD 10: C51-C58), prostate (ICD 10: C60-C63), urinary tract (ICD 10: C64-C68), and lymphoid and hematopoietic tissue (ICD 10: C81-C96)) were performed for both sexes and all age groups combined (Figure 2). To counteract the problem of multiple comparisons as well as due to the large patient samples, IRR values between 0.02 and <0.05 were considered statistical tendency. Multivariable Poisson regression models adjusted for diabetes mellitus, thyroid gland disorders, obesity, diseases of esophagus, stomach and duodenum, diseases of liver, chronic bronchitis and COPD. Analyses were carried out using SAS version 9.4 (SAS institute, Cary, NC, USA).

## 3. Results

### 3.1. Basic Characteristics of the Study Sample

The present study included 12,095 patients with infectious mononucleosis and 12,095 matched individuals without infectious mononucleosis. The basic characteristics of the study cohort are shown in Table 1. There was an average age of 31 years, with 76% of the patients being between 14 and 40 years old. Females accounted for 58.6% of the patients. During the follow-up period, patients visited their primary care physician an average of 5.4 times per year. Patients with infectious mononucleosis showed significantly higher prevalence of thyroid and liver diseases, but significantly lower prevalence of diabetes mellitus and obesity (Table 2).

### 3.2. Association between Infectious Mononucleosis and the Development of Cancer

In patients diagnosed with infectious mononucleosis, the overall cancer incidence was 5.3 cases per 1000 person years versus 4.4 cases per 1000 person years in patients without mononucleosis (Table 2). In multivariable regression analyses, infectious mononucleosis showed a non-significant trend towards a higher incidence of cancer (incidence rate ratio (IRR): 1.17, 95% CI: 1.00–1.36, *p* = 0.044). This non-significant association was only observed in the age group > 50 years (IRR: 1.32; 95% CI: 1.04–1.67, *p* = 0.021) and among men (IRR: 1.36; 95% CI: 1.07–1.72, 0.011, Table 2). 

### 3.3. Association between Infectious Mononucleosis and Different Cancer Entities 

The IRRs are displayed in Figure 2. Infectious mononucleosis was significantly associated with an increased incidence of tumors of hematopoietic and lymphoid tissues (IRR: 1.75; 95% CI: 1.22–2.50, *p* = 0.002). A strong association was noted for infectious mononucleosis and prostate cancer (IRR: 3.09; 95% CI: 1.23–7.76, *p* = 0.016) but this association did not reach the corrected *p*-value of <0.01 as a benchmark of statistical significance. There was no trend towards an association between infectious mononucleosis and cancer of the lip, oral cavity and pharynx, female genital organs, breast, skin, urinary tract or digestive organs (Figure 2). Finally, we observed a non-significant trend towards a negative association between infectious mononucleosis and respiratory organ cancer (IRR: 0.57; 95% CI: 0.29–1.09, *p* = 0.088, Figure 2).

## 4. Discussion

In this study, we investigated a large cohort of more than 24,000 outpatients in Germany for a possible association between infectious mononucleosis, representing the clinical manifestation of EBV infection, and cancer risk using data from a representative real-world database. According to our findings, infectious mononucleosis is significantly associated with an increased incidence of tumors of hematopoietic and lymphoid tissues. In contrast, no significant association was found for the overall risk of cancer in patients with infectious mononucleosis compared to the matched non-mononucleosis cohort, but there was still a tendency for a higher incidence of cancers in infectious mononucleosis patients in the age group > 50 years and among men. 

Infectious mononucleosis has been associated with the development of Hodgkin’s lymphoma for over 30 years, suggesting a causal role of EBV infection [10,11,12,13,14,15,16,17,18]. Interestingly, studies identified the presence of EBV in the characteristic Sternberg-Reed cells of Hodgkin’s disease in about 40–50% of cases [19,20,21,22]. A UK case–control study demonstrated an Odds ratio of 9.16 (95% CI = 1.07–78.31) for EBV-positive Hodgkin lymphoma after infectious mononucleosis [12]. Similarly, Hjalgrim and colleagues, who examined cancer incidence in a population-based cohort from Denmark and Sweden of more than 38,000 patients with infectious mononucleosis, found that patients with infectious mononucleosis were at increased risk for Hodgkin’s disease. It was concluded that the relative risk of Hodgkin’s disease will remain elevated more than twofold for up to 20 years after the diagnosis of infectious mononucleosis [11]. However, apart from a moderately increased risk of skin cancer, the authors found no increased cancer risk associated with infectious mononucleosis for cancers other than Hodgkin’s disease. This observation is consistent with the results of our study, which also revealed no significantly increased overall cancer risk associated with infectious mononucleosis. 

To our knowledge, our study was further the first to indicate a strong trend toward an increased risk of prostate cancer associated with infectious mononucleosis, although the corrected level of statistical significance was not reached. The role of EBV in the development and progression of prostate cancer has remained controversial in the literature; however, EBV is reported to be detectable in 40–50% of prostate cancer specimens [23]. It is suggested that EBV infection with subsequent expression of EBV latency genes could enhance survival of premalignant cells and provide anti-apoptotic properties to epithelial cells in the premalignant stage [24]. At the same time, changes such as inflammation in stromal parts of the tissue could lead to modulation of EBV latency gene expression, altering the growth characteristics of the tissue and triggering carcinogenesis [25].

In contrast, for respiratory organ cancer, we found a non-significant trend toward a negative association with infectious mononucleosis. Indeed, there are no reports in the literature to date of a possible association between infectious mononucleosis and respiratory organ cancer. A relationship between respiratory organ cancer and EBV, however, has already been investigated in several studies [26,27]. Epidemiologic evidence suggests that EBV may play a role in a very limited subgroup of lung cancer [26,27]. In this context, pulmonary lymphoepithelioma-like carcinoma (PLELC), which is a rare form of non-small cell lung cancer closely associated with EBV infection, has to be mentioned [27]. One possible explanatory approach for a potentially negative association between infectious mononucleosis and respiratory organ cancer could be the life cycle of EBV, which occurs mainly in oral epithelial cells and in B cells [27]. In contrast to the well-established pathway by which EBV enters B cells, it is still largely unclear how EBV is able to invade other epithelia such as those of the lung [26,27]. Possibly, demographic aspects could play a role here, supported by the fact that EBV-positive lung carcinomas are predominantly found in Asian populations [27].

Another finding of our study was a significantly higher prevalence of thyroid and liver disease in patients with infectious mononucleosis. There are few case reports in the literature that indicated an association between infectious mononucleosis and the occurrence of thyroid disease [28]. However, a possible etiologic role of EBV itself has been discussed, particularly for autoimmune thyroid diseases. In this context, nuclear expression of EBV non-coding RNA EBER (EBV-encoded RNA) was detected in 80.7% of Hashimoto’s thyroiditis and in 62.5% of Graves’ disease cases [29]. Interestingly, it is already known that infectious mononucleosis is associated with other immunological diseases, with evidence for multiple sclerosis in particular [9,30]. 

With respect to liver involvement, transient elevation of serum aminotransferase levels was noted in up to 90% of young adults with infectious mononucleosis, with older adults having higher liver values than adolescents [31,32]. However, chronic liver damage following infectious mononucleosis has not been described [33]. 

Infectious mononucleosis occurs in 25–70% of cases in the Western population during initial infection with EBV [34] and has a peak between the ages of 15 and 25 years [35]. Once infected, EBV persists in the organism life-long [34]. While approximately 30% have seroconversion for EBV by the age of 14, up to 95% are infected with the virus by the age of 40 [35]. Although almost the entire world population is infected with EBV, the virus seems to cause very different diseases in different regions of the world [36,37,38]. The majority of EBV carriers do not present with clinical symptoms due to viral infection. However, EBV is etiologically associated with the development of different cancers, including several types of B-, T-, and NK-cell lymphomas, as well as carcinomas of the nasopharynx, stomach, parotid gland, and thymus [36,38,39]. 

Globally, about 200,000 annual cancer cases and approximately 2% of all cancer-related deaths are thought to be EBV-associated [27,40,41]. Interestingly, there is a very heterogeneous prevalence of EBV-associated malignancies worldwide. Nasopharyngeal carcinoma, for example, is more common in Southeast Asia, where it accounts for the most common tumor in young adults, as well as in North Africa, whereas it is rare in Europe. While Burkitt’s lymphoma is endemic in central Africa, T-cell lymphomas are almost exclusively described in Japan [36]. The existence of different viral strains or subtypes found only in limited geographic areas, as well as deletions or polymorphisms in the EBV genome, have been discussed in the literature as possible explanations for the differences in disease pattern [42,43]. Delecluse and coauthors identified that virus type M81, isolated from a nasopharyngeal carcinoma, infiltrates epithelial cells of the nasal mucosa in addition to immune system B cells [39]. In contrast, virus types associated with causing infectious mononucleosis almost exclusively infect B cells. 

Age of onset of EBV infection, environmental factors such as nitrosamines or smoking are hypothesized to have an impact on cancer risk. In addition, cofactors such as concurrent infection with Plasmodium falciparum in the case of Burkitt’s lymphoma or immunosuppression in EBV associated B-cell lymphoma or post-transplant lymphoma (PTLD) have been attributed an important role [44,45]. Additionally, genetic determinants are discussed as contributing factors [46]. 

EBV has been shown to affect various cell types, including B cells, epithelial cells, and T cells [47]. Several glycoproteins of the virus are involved in enabling the virus to invade the host cell. The envelope protein gp350/gp220 is thought to be of major importance in this process [48]. It has a high affinity for the complement receptor type 2 CD21 (CR2), which is mainly expressed on B lymphocytes [48,49]. The potential of EBV to immortalize B cells is thought to play an important role in linking the virus to the pathogenesis of lymphoproliferative disorders. In this context, the expression of the six EBV nuclear antigens (EBNA1–6) and the two latent membrane protein genes (LMP1 and LMP2) have been found to be of relevance. The cellular tumor necrosis factor (TNF)-receptor-1-associated death domain protein (TRADD), which mediates the induction of apoptosis as well as the activation of nuclear factor kappa B (NF-κB) by cellular tumor necrosis factor receptor 1 (TNFR1), is thought to be a crucial signaling mediator of LMP1 and critically involved in the immortalization process [50]. Furthermore, EBV lytic replication has been identified as a cancer risk factor [51,52]. 

After all, the property of EBV gene products to be involved in cell proliferation as well as apoptosis and immunological processes may provide an explanation for why the virus is associated with the development of so many different diseases. [53]. 

Our study’s strengths are its population-based setting, cohort size, long follow-up period, as well as the use of a database that has already been proven to be representative for Germany [7]. Nevertheless, there are also limitations to our study, in part because of the study design. For example, diagnoses are based on documentation of ICD-10 codes by general practitioners. Therefore, we cannot exclude the possibility that diagnoses were sometimes misclassified or that the coding was missing from the ICD-10 coding system. In this line of thinking, it is important to note that our study includes only patients with a documented diagnosis of infectious mononucleosis. Thus, patients with asymptomatic EBV infection, which did not lead to a medical consultation, are not included and the non-mononucleosis cohort might include a proportion of patients with clinically inapparent EBV infection vice versa. Nevertheless, the study results of Hjilgram and colleagues indicated that the accuracy of the diagnosis of infectious mononucleosis did not differ significantly when evidence of acute infectious mononucleosis infection was established either serologically or clinically [11]. Moreover, our study explicitly examined the association between cancer and infectious mononucleosis but not EBV. As such, it is not feasible to draw any conclusions about EBV strains or subtypes. Furthermore, tumors of hematopoietic and lymphoid tissues could not be subdivided in more detail because of the existing ICD coding. Finally, the German Disease Analyzer database does not contain information on patients’ lifestyle or socioeconomic status, risk factors such as smoking, immune status, immunological markers, or genetic factors that would have allowed further analyses. 

## 5. Conclusions

In summary, we observed a significant association between infectious mononucleosis and an increased incidence of tumors of hematopoietic and lymphoid tissues in a large real-world cohort of outpatients in Germany. Moreover, we describe a non-significant trend towards an increased incidence of prostate cancer among males with infectious mononucleosis. Further clinical as well as experimental studies are needed to dissect the potential pathophysiological involvement of EBV in human carcinogenesis, which might lead to further advances in vaccine research efforts to develop a preventive agent against certain oncogenic EBV variants.

## Figures and Tables

**Figure 1 cancers-14-05837-f001:**
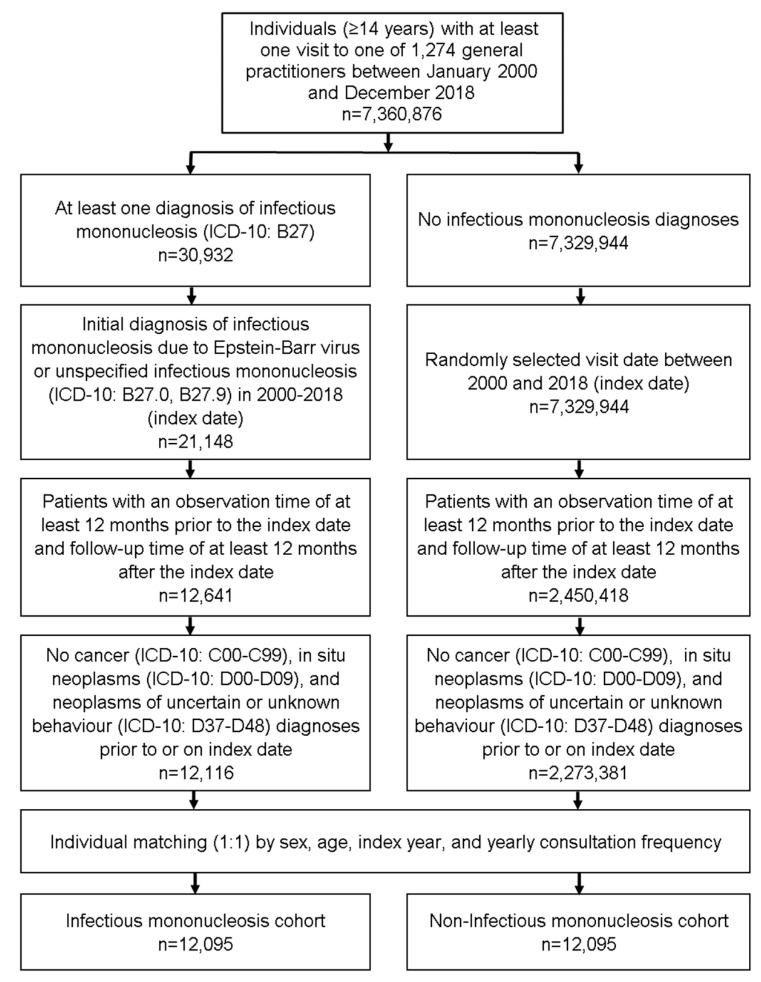
Selection of study patients.

**Figure 2 cancers-14-05837-f002:**
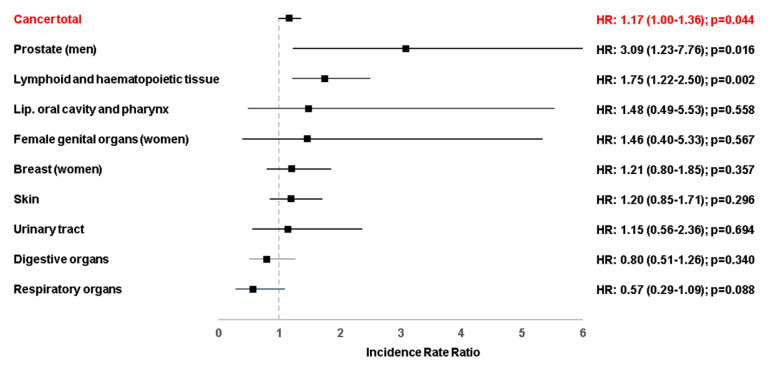
Association between infectious mononucleosis and different cancer diagnoses in patients followed in general practices in Germany (Multivariable Poisson regression models). Multivariable Poisson regression models adjusted for diabetes mellitus, thyroid gland disorders, obesity, diseases of esophagus, stomach and duodenum, diseases of liver, chronic bronchitis and COPD.

**Table 1 cancers-14-05837-t001:** Association between infectious mononucleosis and the subsequent cancer diagnoses in patients followed in general practices in Germany (Multivariable Poisson regression model).

Cohorts	Incidence among Individuals with Infectious Mononucleosis (Cases per 1000 Patient Years)	Incidence among Individuals without Infectious Mononucleosis (Cases per 1000 Patient Years)	Incidence Rate Ratios (95% CI) *	*p*-Values
Total	5.3	4.4	1.17 (1.00–1.36)	0.044
Age 14–20	1.1	1.2	0.96 (0.55–1.66)	0.882
Age 21–30	2.2	2.6	0.83 (0.54–1.28)	0.393
Age 31–40	4.3	3.4	1.25 (0.83–1.90)	0.291
Age 41–50	8.7	6.9	1.22 (0.88–1.69)	0.231
Age > 50	19.6	14.5	1.32 (1.04–1.67)	0.021
Women	4.1	4.6	1.04 (0.86–1.27)	0.681
Men	6.9	4.1	1.36 (1.07–1.72)	0.011

* Adjusted for diabetes mellitus, thyroid gland disorders, obesity, diseases of esophagus, stomach and duodenum, diseases of liver, chronic bronchitis, and COPD.

**Table 2 cancers-14-05837-t002:** Basic characteristics of the study sample (after 1:1 matching by propensity scores based on sex, age, index year, and yearly consultation frequency).

Variable	Proportion Affected among Individuals with Infectious Mononucleosis (%) *n* = 12,095	Proportion Affected among Individuals without Infectious Mononucleosis (%) *n* = 12,095	*p*-Value
Age (Mean, SD)	31.2 (14.6)	31.3 (14.7)	0.400
Age 14–20	32.4	30.9	0.121
Age 21–30	25.4	26.4	
Age 31–40	17.2	16.9	
Age 41–50	12.8	13.5	
Age >50	12.2	12.3	
Women	58.6	58.6	1.000
Men	41.4	41.4	
Yearly consultation frequency	5.4 (4.5)	5.4 (4.5)	1.000
Diagnoses documented within 12 months prior to the index date			
Diabetes mellitus	3.0	4.8	<0.001
Obesity	5.4	7.8	<0.001
Thyroid gland disorders	17.6	14.9	<0.001
Diseases of esophagus, stomach and duodenum	22.6	21.7	0.101
Diseases of liver	8.9	4.4	<0.001
Chronic bronchitis and COPD	6.86	6.84	0.980

Proportions of patients given in %, unless otherwise indicated. SD: standard deviation.

## Data Availability

The datasets used and analyzed during the current study are available from the corresponding author on reasonable request.

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
