# Peer review of "The Association between Infectious Mononucleosis and Cancer: A Cohort Study of 24,190 Outpatients in Germany"

_cancers, 2022, doi:10.3390/cancers14235837_

Round 1
Reviewer 1 Report
This is a well-written paper by Christoph Roderburg and colleagues aimed at retrospective cohort study consisting of a large sample of outpatients in Germany with a diagnosis of infectious mononucleosis. Author collected data from German Disease Analyzer database (IQVIA), which contained drug prescriptions, diagnoses, and basic demographic data obtained directly and in anonymous format from computer systems of general practitioners and specialists in Germany. The database covers approximately 3% of all outpatient practices. Authors performed strict matching between patients with infectious mononucleosis and patients without infectious mononucleosis using propensity scores on the basis of sex, age, index year, and annual consultation frequency (see Figure 1). These study strengths are its population-based setting, cohort size (24 000 subjects) and long follow-up period (2000-2018). Accordingly, the authors provide the evidence on a significant association between infectious mononucleosis and an increased incidence of tumors of the hematopoietic and lymphoid tissues in a large real-world cohort of outpatients in Germany. Moreover, they describe a non-significant trend towards an increased incidence of prostate cancer among males with infectious mononucleosis. Authors extensively discussed latest literature data and their own experimental results. This is an interesting study, and clinically valuable, especially for those clinicians who provide medical care in Germany population. This manuscript provides comprehensive information on the potential role of EBV in carcinogenesis issues.
Minor issue:
1. Although manuscript may lack some originality, the unique and rigorous methodological approach deserves its recognition and promotion in high impact journal for readers and clinicians.
Taken together, paper by Roderburg and colleagues represent a worthwhile contribution to the cancer research. I recommend the manuscript for further publication process.
Reviewer 2 Report
In my opinion the paper can be informative and provide a valuable source document for anyone requiring a primer to know and understand this issue. But, numerous shortcomings in the section Methods, Results and Discussion make this paper not appropriate for publication in this form and significant corrections should be made (major revision). Additionally, one of the reasons for this is the lack of data about Ethical considerations. Some comments:
- Lines 51-54: Check the cited references and correct in a way that you are citing the appropriate original reference, i.e. the appropriate IARC monograph about carcinogens in humans, since the year 1997 is stated.
- Line 61: To improve readability, add subsections: Study setting, Study design, Study sample, Simple size calculation.
- Lines 62-69: State whether the Data source was publicly available, or whether a permission to access and use data from this database was necessary.
- Line 72: Inscribe the `Case definition` that was used for the diagnosis of infectious mononucleosis during the study period, with citation of an appropriate reference. State whether the diagnostic criteria for infectious mononucleosis have changed during the study period.
- Line 79: State whether individual matching for age was done without or with a deviation for years, e.g. +/- 2 years, +/- 5 years or similar.
- Line 95: State how collinearity between all variables was assessed in this study (state by which test, state the results of the estimated collinearity, state how the issue of collinearity was handled in this manuscript).
- Lines 96-97: State for which variables and by what criteria was adjusting done.
- Lines 97-98: In the section Results the results of analyses are not presented as it was stated here `These models were performed separately for women and men as well as five age groups.`. Therefore, all results (especially on Figure 2) should be presented `separately` by sex and age groups, as stated in the Methods section.
- Lines 98-102: State `Lip, oral cavity and pharinx` and Total cancers, with the corresponding ICD codes, as stated in Figure 2.
- Lines 102-106: Define the indicators that will be presented in the paper: IRR, 95%CI, HR, etc.
- Lines 174-178: Discuss the findings regarding prostate cancer in detail. Compare to results of other studies.
- Lines 178-233: Reconstruct the entire text, following as you have yourself stated and underlined that in this manuscript you are presenting data about infectious mononucleosis, and not EBV.
- Line 233: Add the discussion of presented results, whether or not they were statistically significant. Pay particular attention in the section Discussion to the result for cancers of `Respiratory organs`, with an interpretation of the results.
- Lines 212-214: Cite the appropriate reference.
- Lines 234-254: The paragraph about Strengths and limitations of the study is written very well.
- Line 267: Check and correct whether the `Consent to publication` refers to authors or to participants in this study.
Round 2
Reviewer 2 Report
Thank you for the opportunity to re-review the manuscript ID: cancers-2022737. The authors have addressed most, but not all, of the issues highlighted in my review. Thank you to the authors for their responses to my comments. I believe that the changes they have made have significantly improved the manuscript. But, some corrections are necessary.
Some comments:
- The answers to the first 3 comments from the previous review are missing (Lines 51-54: Check the cited references and correct in a way that you are citing the appropriate original reference, i.e. the appropriate IARC monograph about carcinogens in humans, since the year 1997 is stated.
Line 61: To improve readability, add subsections: Study setting, Study design, Study sample, Simple size calculation.
Lines 62-69: State whether the Data source was publicly available, or whether a permission to access and use data from this database was necessary.).
The missing responses need to be provided.
- Enter in the revised manuscript the answers under point 1 and 3 from the `Point-by-point response' document. Apply to all comments.
- Regarding the response - point 5 from the `Point-by-point response' document --- (`5. Lines 97-98: In the section Results the results of analyses are not presented as it was stated here `These models were performed separately for women and men as well as five age groups. Therefore, all results (especially on Figure 2) should be presented `separately` by sex and age groups, as stated in the Methods section.
Response: We intentionally described age- and sex stratified regression prior to cancer site specific analyses in the Methods. We planned and conducted as well as displayed age- and sex specific analyses for the association between mononucleosis and cancer in total. Else these would be too many analyses, too many models and too big risk of “accidental discoveries”.`), --- it is necessary for the authors to write in the Methods section only what will be presented in Results.
It is an issue when described analyses are not conducted nor or presented in the manuscript.
Round 3
Reviewer 2 Report
The Authors have made the necessary corrections.
I would like to point out that it is not clear how the Authors did not receive a complete review of mine in the first round of review, since that is what they stated in their response - "First of all, please excuse us for not addressing the first 3 points. We must have used an initial incomplete version of your review before, so we missed the points mentioned.", while my full report is in the system.